# Salivary Biomarkers in Patients with Sjögren’s Syndrome—A Systematic Review

**DOI:** 10.3390/ijms222312903

**Published:** 2021-11-29

**Authors:** Ju-Yang Jung, Ji-Won Kim, Hyoun-Ah Kim, Chang-Hee Suh

**Affiliations:** Department of Rheumatology, Ajou University School of Medicine, Suwon 16499, Korea; serinne20@hanmail.net (J.-Y.J.); jwk722@naver.com (J.-W.K.); nakhada@naver.com (H.-A.K.)

**Keywords:** Sjögren’s syndrome, saliva, protein, biomarker

## Abstract

Sjögren’s syndrome (SS) is a chronic autoimmune disease characterized by dry mouth and dry eyes, with lymphocytic infiltration of the exocrine glands. Saliva is becoming a useful tool to determine the clinical and pathological characteristics of SS because the collection method is easy and non-invasive. Since 1900, salivary proteomic analysis has been performed continuously using a variety of optimized analytical methods. Many studies have identified distinct characteristics of salivary proteins in patients with primary SS, and the changes were related to chronic inflammation and overproduction of immunoglobulins or downregulated secretory function. Several proteomic studies using whole or parotid saliva have evaluated whether several salivary proteins can be used to discriminate SS, including salivary β2-microglobulin, calprotectin, carbonic anhydrase VI, neutrophil gelatinase-associated lipocalin, sialic acid-binding immunoglobulin-like lectin-5, and tripartite motif-containing protein 29. In addition, salivary proinflammatory cytokine levels have been reported to be increased in patients with SS. Although these candidate salivary proteins have exhibited considerable differences in patients with SS, more data are needed to confirm their role as biomarkers. Moreover, the identification of salivary characteristics that can accurately reflect disease activity, predict treatment response and prognosis, and diagnose SS is anticipated.

## 1. Introduction

Sjögren’s syndrome (SS) is a chronic autoimmune disease characterized by dry mouth and dry eyes, is caused by chronic lymphocytic infiltration of the exocrine glands, mainly the salivary and lacrimal glands [1,2]. Although xerostomia and dry eye(s) are typical manifestations, arthritis, parotid gland enlargement, interstitial lung disease, and lymphadenopathy can develop in 30–50% of patients with SS. Patients with SS can experience complications including atrophy of the tongue papillae, periodontal disease, abnormal taste sensation, oral ulcers, and alterations in voice or taste. SS can occur in those with chronic autoimmune diseases, such as rheumatoid arthritis (RA) and systemic sclerosis and when it occurs without these comorbid diseases, it is defined as primary SS (pSS). Patients with SS have a risk for non-Hodgkin lymphoma and mucosa-associated lymphoid tissue (MALT) lymphoma. The major pathogenic mechanisms include B cell hyperactivity and the production of autoantibodies against Ro/SS-A and La/SS-B [3]. Pathological findings, which are identified using labial minor salivary gland biopsy, include lymphocytes composed of B and T cells infiltrating the glands in inflammatory lesions, especially CD4-positive cells [4]. In addition, increased proinflammatory cytokines or chemokines mediating the recruitment and differentiation of such lymphocytes contribute to inflammation of the glands, leading to germinal center formation.

Saliva is a complex biological fluid secreted from the major and minor salivary glands and is composed of water, various molecules from the blood, and salivary proteins from the oral cavity [5]. Abundant enzymes, hormones, antibodies, antimicrobial molecules, and growth factors in the blood move into saliva; as such components of saliva are similar to serum, which reflects the physiological state of the body, including hormonal, nutritional, and metabolic variations. Although the concentrations of most substances are lower than levels in the blood, the use of saliva as a diagnostic tool has clinical advantages. Compared with blood samples, collecting saliva is simple and non-invasive; as such, there are no side effects, such as needle pain or bleeding. In addition, chronic autoimmune diseases, such as SS, require regular monitoring to evaluate disease activity and modulate treatment, for which repeated tests are necessary. In such cases, collecting saliva is preferred over other sample types.

Newly developed and more sensitive technologies, including molecular diagnostics and nanotechnology, have been used to determine minimal differences in saliva among different conditions. Several studies have identified the role of saliva in differentiating, monitoring, and predicting prognosis in various conditions including infection, malignancy, and drug use. Because saliva is produced in the vicinity of the oral cavity and released into the mouth, the clinical use of saliva is reasonable in oral diseases. For example, the levels of proinflammatory cytokines, including interleukin (IL)-6, IL-8, tumor necrosis factor (TNF)-α, and IL-1β in saliva, have been reported to be elevated in patients with oral squamous cell carcinoma, thus supporting their potential utility as diagnostic biomarkers [6].

There are two methods for collecting saliva—one uses stimuli, the other does not. Paraffin wax or chewing gum are used as stimuli for collecting saliva through induction of masticatory action, leading to an increased salivary flow rate. Effectively collecting saliva in patients with SS is difficult because the saliva flow rate is significantly decreased in those with the disease [7,8]. Therefore, a considerable length of time is required to collect an appropriate amount of saliva, which can negatively affect the quality of the sample [8]. As such, many studies investigating SS have used stimulated saliva as a sample, although such stimuli can affect the quantity and pH of saliva.

Many studies have attempted to identify and evaluate differences in saliva as biomarkers for the diagnosis or monitoring of SS. There are three commonly used classification criteria, made by the American-European Consensus Group (AECG) in 2002, the Sjögren’s International Collaborative Clinical Alliance (SICCA) in 2012, and the American College of Rheumatology/European league against rheumatism (ACR/EULAR) classification criteria in 2016 (Table 1) [2,9,10]. In many studies looking for salivary biomarkers for SS, 2002 AECG criteria have been used, and 2012 SICCA and 2016 ACR/EULAR criteria are used in recent studies. In the present review, we summarize studies analyzing the clinical use of saliva in patients with SS and evaluate whether a characteristic of this bodily fluid can be used as a biomarker for SS.

## 2. Methods

The search strategy adhered to the Preferred Reporting Items for Systematic reviews and Meta-Analyses (i.e., PRISMA) guidelines (Figure 1) [11]. The literature search of the MEDLINE databases used key terms related to ‘‘Sjögren’s syndrome’’ AND the key terms ‘‘saliva.” Several relevant keywords, including “biomarker,” “salivary proteomic,” and “salivary protein,” were used in different combinations for the manual search. The reference lists of the retrieved articles were reviewed to enhance the sensitivity of the search strategy. Potentially eligible candidate articles were screened through matching title and abstract; the entire article was reviewed if it fulfilled the inclusion criterion. In addition, the authors also reviewed and searched the reference lists of the selected articles according to the inclusion criterion, which was, more specifically, English language publication in a peer-reviewed journal. Some articles were excluded after a review of the abstract or the full text if it was irrelevant to the topic in question. The authors independently searched and reviewed the articles and selected references based on the inclusion and exclusion criteria.

## 3. Proteomic Analysis

Proteomic analysis has been used to screen the changes in salivary protein composition in patients with SS, with altered proteins anticipated to be biomarkers for SS. In this process, newly developed and more precise and accurate experimental techniques have been used.

Ryu et al. performed a proteomic analysis using surface-enhanced laser desorption/ionization time-of-flight mass spectrometry (MS) and two-dimensional difference gel electrophoresis (2-DE) and revealed that the levels of β2-microglobulin (β2m), lactoferrin, immunoglobulin κ light chain, polymeric immunoglobulin (Ig) receptor, lysozyme C, and cystatin C were elevated in the parotid saliva of SS patients compared to healthy controls (HCs) [12]. Moreover, levels of β2m and lactoferrin were increased three to four-fold in the parotid saliva of SS patients, with low to high focus compared to those of the non-SS group according to enzyme-linked immunosorbent assay (ELISA). The results revealed increased levels of inflammatory proteins and decreased levels of acinar proteins in the saliva of patients with SS.

A proteome analysis of whole saliva using 2-DE revealed 16 proteins, that exhibited higher or lower expression in 24 patients with pSS compared to those in 16 HCs, and 10 proteins that were detected in only one group of patients with pSS and HCs [13]. The levels of salivary fatty acid-binding protein, β-actin, β-actin fragment, leukocyte elastase inhibitor, and glutathione S-transferase were increased in those with SS, and salivary α-amylase precursor, cystatin precursor, keratin6-L, and prolactin-inducible protein precursor (PIP) were decreased. In addition, salivary cystatin D and precursors, and carbonic anhydrase (CA)-VI were detected only in HCs, while salivary calgranulin B, cyclophilin A, lipocalin-1 precursor, phosphatidylethanolamine binding protein, immunoglobulin-C protein, zinc-α2-glycoprotein precursor were detected only in patients with pSS. Such changes indicate acute and chronic inflammation in those with SS. Salivary PIP plays a role in the formation of the enamel pellicle and mucosal defense, suggesting chronic inflammation of the oral environment in patients with pSS, such as chronic blepharitis and the influence of increased salivary IL-6 levels in patients with pSS.

A panel of protein and messenger RNAs (mRNAs) was identified in the saliva of patients with pSS using MS and expression microarray analysis [14]. The validation revealed that levels of salivary cathepsin D, α-enolase, and β2m were elevated in patients with pSS compared to those with systemic lupus erythematosus (SLE) and HCs [15]. In addition, mRNA expression of myeloid cell nuclear differentiation antigen, low-affinity IIIb receptor for the Fc fragment of IgG, and guanylate binding protein 2 in saliva were elevated in pSS compared with HCs and patients with SLE. Other proteomic analyses compared salivary candidate proteins among pSS, secondary SS (sSS) and other sicca patients [16]. While salivary α-enolase (59KDa) level was elevated in non-SS sicca patients and systemic sclerosis-sSS patients, salivary β2m and immunoglobulin k light chain were elevated in those with pSS compared to HCs or RA-sSS according to Western blotting results.

Proteomics using Ingenuity Pathway Analysis (IPA) of the parotid saliva of patients with SS revealed 472 proteins identified only in those with pSS, 57 proteins identified only in HCs (*n* = 57), and 206 proteins upregulated or 34 downregulated in patients with pSS [17]. These proteins include salivary exosomes and small membrane-bound vesicles in the saliva that modulate T cell activation and are involved in antigen presentation.

A 187-plex capture antibody-based assay was used to identify salivary biomarkers for pSS, and changes in 61 proteins among 48 patients with SS and 24 non-SS subjects (12 RA and 12 HCs) [18]. The multiple-analyte profile (MAP) produced a discriminant function consisting of clusterin, IL-5, fibroblast growth factor 4, and IL-4, with accurate group prediction for 93.8% of patients with SS and correct identification of 100% of non-SS subjects. Altered salivary proteins in patients with SS are associated with immune response, immune cell differentiation, and tissue homeostasis.

Deutsch et al. performed a proteomic analysis using quantitative dimethylation liquid chromatography-tandem MS after depleting amylase and IgG and revealed 79 proteins that differed in expression between patients with SS and HCs [19]. These proteins are involved in the defense response, regulation of apoptosis, stress response, and cell motion.

Proteomic analysis using isobaric mass tagging (iTRAQ) and lectin affinity capture MS revealed that many proteins in parotid and whole saliva were expressed differently in patients with SS compared to those with non-SS sicca symptoms and HCs [20]. The validation of candidate proteins by immunoblotting revealed that β2m in parotid saliva was upregulated in five patients with SS compared to five HCs, CA-VI, and bactericidal/permeability increasing fold-containing family B2 in whole saliva were downregulated in five patients with SS compared in five HCs.

Salivary and tear proteomic analysis using liquid chromatography (LC)-MS revealed upregulated proteins, including neutrophil gelatinase-associated lipocalin (NGAL), granulin, calmodulin, epididymal secretory protein-1, and calmodulin-like protein 5 in 27 patients with pSS compared to those in 32 HCs [21]. These proteins are associated with immunity, cell signaling, and wound repair. Moreover, the Database for Annotation, Visualization, and Integrated Discovery (DAVID) demonstrated enhanced pathways of adaptive immune response and cellular component assembly for saliva extracellular vesicles (EV). In addition, Aqrawi et al. performed proteomic analysis using LC-MS to determine an association between altered salivary, tear, and EV proteins and histopathological characterization of patients with pSS [22]. Upregulated proteins in stimulated whole saliva of patients with pSS were peptidyl-prolyl cis-trans isomerase FK506-binding protein 1A, CD44, β2m, secreted Ly-6/uPAR-related protein 1, and clusterin. Upregulated proteins in EVs isolated from stimulated whole saliva of patients with pSS included CD44, major vault protein, NGAL, ficolin-1, and annexin A4.

Proteomic analysis of saliva, plasma, and salivary gland tissue from SS patients using LC-MS was performed [23]. Principle component analysis using each sample revealed that salivary proteins involved in complement and coagulation cascades were able to discriminate patients with pSS, and proteins that are known to be associated with salivary secretion were found less frequently in patients with pSS. Interestingly, saliva data demonstrated a significant difference in the protein expression profiles of patients with pSS and non-pSS patients. Forty proteins in stimulated whole saliva differed between the 24 patients with pSS and 16 non-SS controls. Neutrophil elastase, calreticulin, tripartite motif-containing protein (TRIM) 29, clusterin, and vitronectin were upregulated, and histatins 1 and 2, basic salivary proline-rich proteins (PRPs) 1, 2, and 4 were downregulated in stimulated whole saliva. In addition, they used salivary TRIM29 as a biomarker for pSS, and the area under the curve (AUC) of the combination of salivary TRIM29 and serum anti-SSA/Ro was 0.995 [24]. TRIM proteins are involved in pathogen recognition and regulation of transcriptional pathways in host defense, and TRIM29 has been shown to inhibit innate immune activation in viral infections [25]. Salivary TRIM29 needs to be evaluated for clinical applicability as a biomarker with its biological functions in pSS.

Proteomic analysis using an experimental SS mouse model revealed that salivary C3, complement factor H (CFH), serpin family G member 1 (SERPING1), fibrinogen alpha (FGA), and fibrinogen gamma (FGG) expressions were different between SS model and control mice [26]. Downregulation of salivary C3, CFH, SERPING1, FGA, FGG is associated with activation of the alternative complement system and defects in the complement system with low production of complement proteins.

Immune complex formation and deposition are critical in the pathogenesis of autoimmune disease, including SS, and the presence of immune complexes in the serum, blood vessels, and glomeruli has been found in patients with SS. Proteomic analysis using immune complex-capturing beads and nano-LC-MS/MS identified 998 immune complex-antigen present in the saliva of nine patients with SS, but not in the saliva of seven non-SS patients [26]. Neutrophil defensin 1 and small proline-rich protein 2D were found most frequently in patients with SS.

## 4. Salivary Biomarkers of Sjogren’s Syndrome

### 4.1. Salivary β2-Microglobulin (β2m)

β2m is a low-molecular-weight protein of the major histocompatibility complex-I, which is expressed on antigen presenting cells including T and B lymphocytes and regulated by interferon [27]. Several proteomic analyses mentioned above have demonstrated that salivary β2m levels were increased in patients with SS [16,17] (Table 2). Immunosorbent assay identified that patients with SS had a significantly higher concentration of salivary β2m than HCs or in non-SS-sicca patients did [15,28,29,30,31]. A study showed the higher levels of salivary β2m had an AUC of 0.95, a sensitivity of 0.94, and a specificity of 0.85 in receiver operating characteristic curve analysis for distinguishing and HCs [32]. Elevated salivary β2m is considered to result from lymphocytes activation and infiltration in salivary glands of patients with SS. A study found that salivary β2m measured by ELISA was positively correlated with the EULAR Sjögren’s Syndrome Patient Reported Index (ESSPRI) and histopathological change in 71 patients with pSS, and the AUC of salivary β2m was 0.965 (95% confidence interval 0.891–0.994; *p* < 0.0001) [33]. The concentration of salivary β2m might be a reliable biomarker for differentiating SS and representing disease activity of SS.

### 4.2. Salivary Lactroferrin

Lactoferrin is an iron-binding glycoprotein present in secretions that has anti-microbial properties and is involved in lymphocyte exotaxis [34]. Moreover, the release of surface-expressed lactoferrin from polymorphonuclear neutrophils regulates cytokine production in T-helper cell type 1. Several studies showed that concentrations of lactoferrin in unstimulated whole or parotid saliva was increased in patients with SS compared to those in HCs, and strong lactoferrin staining of the salivary ducts in patients with SS [35,36,37]. Those data indicated that elevation of salivary lactoferrin is associated with lymphocytic infiltration on salivary glands and the immune imbalance that occurs in SS. However, salivary lactoferrin levels had been elevated in other conditions, such as periodontitis and SLE, suggesting a lack of specificity for biomarkers for SS [38,39].

### 4.3. Salivary Neutrophil Gelatinase-Associated Lipocalin (NGAL)

NGAL, also known as lipocalin-2, is one of the lipocalins transporting small and hydrophobic molecules including fatty acids and hormones, and its expression is upregulated in various conditions including infection, diabetes, obesity, and cancer. NGAL was found to modulate innate immunity during infection and regulate the development of autoantibodies to nuclear antigens in a lupus-prone model [40]. NGAL concentrations were upregulated in neutrophils of RA synovium by granulocyte macrophage colony-stimulating factor with synovial proliferative effects [41]. Through proteomic analysis using LC-MS, NGAL expressions were highly elevated in the salivary glands and saliva of patients with SS [21]. Aqrawi et al. revealed that NGAL expression in the acinar epithelium of the salivary gland was detected only in patients with SS, and correlated with the focus score, which represents salivary gland inflammation in SS [42]. In addition, NGAL levels were detected in stimulated whole saliva of 8/11 patients with pSS and 2/11 HCs.

### 4.4. Salivary Soluble Sialic Acid-Binding Immunoglobulin-like Lectin (Siglec)-5

Lee et al. found that the expression of siglec)-5 was elevated in the peripheral blood mononuclear cells of patients with pSS [43]. Siglecs are cell-surface transmembrane receptors on immune cells, modulating the function of immune cells, and have been found to play a role in the defense against infection and autoimmune diseases, including RA and SLE [44]. Siglec-5 is expressed on the surface of neutrophils, monocytes, and macrophages, and its cytoplasmic tail has immunoreceptor tyrosine-based inhibitory motifs, inhibiting cell activation. In a subsequent study, salivary siglec-5 were increased in 170 patients with pSS compared with those in 25 HCs, 78 non-SS sicca patients, and 43 patients with SLE [45]. In addition, the AUC was 0.774 with a sensitivity of 0.69 and a specificity of 0.7. Moreover, pSS patients with higher salivary siglec-5 levels demonstrated a higher xerostomia inventory questionnaire score, serum IgG level, and ocular staining score and a lower unstimulated salivary flow rate and white blood cell count. Soluble siglec-5 might be highly released from the membrane of macrophages or neutrophils in oral mucosa, and the cells depleted of siglec-5 can be activated without inhibitory motifs, leading to the inflammatory condition of SS. However, the mechanism of high concentration of salivary siglec-5 in patients with SS or its pathologic role for SS remained unclear.

### 4.5. Salivary Cytokines

As a chronic autoimmune disease, proinflammatory cytokines play a role in producing the inflammatory response in SS and their levels in saliva have been investigated in patients with SS. While IL-17 and IL-23 were positively stained in the salivary glands of pSS models, IL-17 and IL-6 in saliva were measured at varying levels [46]. In other studies, salivary and serum IL-17A, IL-6, TNF-α, and IL-10 levels were higher in 44 patients with pSS than in 15 HCs, and salivary IL-17A and IL-6 levels were correlated with nitric oxide production, which was associated with infiltration grade progression [47]. In another study, salivary soluble L-selectin and IL-17 levels were increased in 43 patients with pSS compared to those in 31 HCs [48]. Salivary IL-6 and TNF-α levels were higher in 138 patients with pSS than in 100 HCs, while IL-17A and rheumatoid factor-IgA levels were not different [49]. In addition, salivary IL-6 levels were correlated with erythrocyte sedimentation rate and IgG levels, and salivary TNF-α levels were correlated with IgG levels in patients with pSS.

### 4.6. Salivary Autoantibody

Several autoantibodies, including anti-Ro/La, have been found in saliva of patients with SS, which might be associated with inflammation of salivary glands. Anti-Ro/La antibodies in serum are a reliable marker for diagnosis of SS and trigger the autoimmune response in exocrine glands. Hu et al. identified salivary autoantibodies for pSS using immune response protoarrays [50]. Among 24 candidate autoantibodies, anti-histone, anti-transglutaminase, anti-SSA, and anti-SSB antibodies were validated in a cohort of 34 patients with pSS, 34 patients with SLE, and 34 HCs. All four autoantibodies were overexpressed in patients with pSS, and the AUC of anti-histone, anti-transglutaminase, anti-SSA, and anti-SSB antibodies for pSS (versus HCs) were 0.95, 0.87, 0.93, and 0.94, respectively.

Muscarinic type 3 receptor (M3R) is highly expressed in exocrine glands and is known to modulate secretion in salivary acinar cells [51]. Autoantibodies against M3R have been reported to be prevalent in the serum of patients with pSS and related to glandular infiltration, impaired exocrine function, and disease activity. Salivary IgG against M3R was found in 55.8% of patients with pSS, and its positivity was associated with lower age, shorter disease duration, and higher globulin levels in 43 patients with pSS [52]. Anti-M3R autoantibody levels in saliva were increased in 37 patients with pSS compared to that in 26 non-SS-sicca patients and correlated with whole saliva flow rate [53]. The AUC of the anti-M3R antibody for pSS in saliva was 0.84, while in plasma, the AUC was 0.95.

A study that detected tissue-specific antibodies in pSS reported that serum anti-CA-IV and parotid secretory protein (PSP) antibodies were more frequently positive in patients with pSS than in HCs, and salivary anti-CA-IV, salivary protein (SP)-1 and PSP IgG were higher in patients with pSS than in HCs [54]. In several studies, the positivity of these antibodies in serum has been detected in 40–67% of patients with pSS and they are a promising serum biomarkers for pSS.

### 4.7. Salivary Calprotectin

S100 proteins, including S100A8 (myeloid-related protein 8 [MRP-8] or calgranulin A), S100A9 (MRP-14 or calgranulin B), and calprotectin (S100A8/A9), are released from activated phagocytes, activate immune cells, and increase proinflammatory cytokine levels with cytotoxic effects [55]. They modulate the inflammatory response in autoimmune diseases, such as RA and juvenile idiopathic arthritis, and serum calprotectin and S100A12 levels were significantly elevated in patients with pSS compared to reference levels [56]. Serum calprotectin was correlated with anti-Ro and anti-La antibody levels and the fatigue severity scale, while serum S100A12 was correlated with the focus score. In another study, serum calprotectin was increased and correlated with the focus score in patients with pSS [57]. In several studies, salivary S100 protein levels were found to be increased in patients with SS. In addition, salivary calprotectin was elevated in 23 patients with pSS compared with that in 10 HCs and was correlated with the focus score in patients with pSS [58]. In another study, salivary actin cytoplasmic 2, Ig γ-1 chain C region, S100-A8, and S100-A9 were upregulated and selected through proteomic analysis, and S100A8/A9 levels were increased in parotid saliva (*n* = 83), but not in whole saliva (*n* = 56) of patients with SS compared to those of HCs and those with non-SS dry mouth disease [59]. S100A8/A9 levels in whole saliva were increased in SS patients who had been diagnosed with MALT lymphoma compared to those without.

### 4.8. Salivary Carbonic Anhydrase VI

CA-VI is a zinc-containing metalloenzyme that is produced and secreted by the salivary glands. It binds to the enamel pellicle and then catalyzes the conversion of salivary bicarbonate and hydrogen ions to carbon dioxide and water, resulting in the maintenance of salivary pH. Several proteomic analyses have revealed that salivary CA-VI is expressed at a lower level in patients with SS than in HCs [12,14,16]. Downregulated salivary CA-IV in patients with pSS may reduce pH in the acidic oral cavity, thus aggravating a weak condition for caries and other infections.

### 4.9. Salivary Adiponectin

Adiponectin is secreted by adipose tissue, including salivary gland cells, and adenosine deaminase (ADA) is involved in monocyte-to-macrophage differentiation and the development of B and T lymphocytes. In one study, salivary adiponectin and ADA levels were higher in 17 patients with SS and 19 patients with non-SS sicca symptoms compared to those in HCs [60]. The high levels of adiponectin and ADA might be derived from lymphocyte activation and infiltration into the salivary gland. In addition, salivary IL-1β, IL-6, and IL-8 levels were elevated in patients with SS and patients with non-SS sicca symptoms compared to those in HCs. Salivary adiponectin was significantly correlated with salivary IFN-γ, IL-1, IL-8, TNF-α in patients with SS.

## 5. Conclusions

SS is characterized by chronic inflammation, mainly involving the salivary and lacrimal gland, and differences in the composition of saliva arise from pathohistological etiology. Saliva collection is convenient and non-invasive, which is important in the clinical setting. Several proteomic analyses using optimized techniques have been performed, and some alterations in saliva have been reported. Salivary autoantibodies, including anti-M3R antibodies, are less sensitive than anti-Ro/La antibodies, which are currently used. The levels of salivary inflammatory cytokines and calprotectin were increased in patients with SS and were correlated with disease severity or salivary inflammation; however, they demonstrated limited specificity for SS. Although salivary NGAL, siglec-5, and CA-VI have been identified as promising biomarkers for SS, further studies are needed. In addition, these salivary markers, which were different from that of healthy subjects, might be only some part of saliva. Because studies investigating the utility of saliva are now being conducted more continuously, saliva biomarkers may be used to diagnose, monitor treatment response, and predict the prognosis of SS in the future.

## Figures and Tables

**Figure 1 ijms-22-12903-f001:**
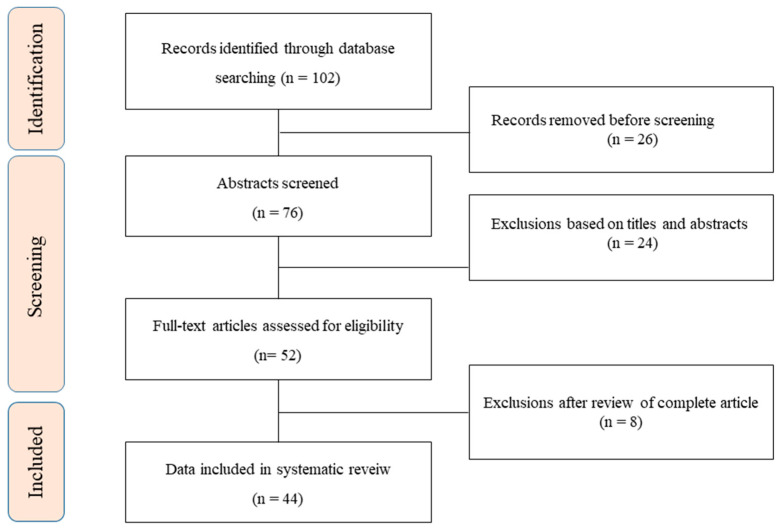
Flow diagram.

**Table 1 ijms-22-12903-t001:** Classification criteria of Sjögren’s syndrome (SS).

	2002 American-European Consensus Group (AECG) [9]	2012 Sjögren’s International Collaborative Clinical Alliance (SICCA) [10]	2016 American College of Rheumatology/European League Against Rheumatism (ACR/EULAR) [2]
Item	I. Ocular symptoms: positive response ≥ 1 of the following questions: 1. Have you had daily, persistent, troublesome dry eyes for more than 3 months? 2. Do you have a recurrent sensation of sand or gravel in the eyes? 3. Do you use tear substitutes more than three times a day? II. Oral symptoms: positive response ≥ 1 of the following questions: 1. Have you had a daily feeling of dry mouth for more than 3 months? 2. Have you had recurrently or persistently swollen salivary glands as an adult? 3. Do you frequently drink liquids to aid in swallowing dry food? III. Ocular signs (objective evidence of ocular involvement): positive result ≥ 1 of the following two tests: 1. Schirmer’s I test, performed without anesthesia (<5 mm in 5 min) 2. Rose bengal score or other ocular dye scores (>4 according to van Bijsterveld’s scoring system) IV. Histopathology: In minor salivary glands, focal lymphocytic sialoadenitis, evaluated by an expert histopathologist, with a focus score >1, defined as a number of lymphocytic foci (which are adjacent to normal-appearing mucous acini and contain >50 lymphocytes) per 4 mm^2^ of glandular tissue V. Salivary gland involvement (objective evidence of salivary gland involvement): positive result ≥ 1 of the following diagnostic tests: 1. Unstimulated whole salivary flow (<1.5 mL in 15 min) 2. Parotid sialography showing the presence of diffuse sialectasias (punctate, cavitary, or destructive pattern), without evidence ofobstruction in the major ducts 3. Salivary scintigraphy showing delayed uptake, reduced concentration and/or delayed excretion of tracer VI. Autoantibodies: presence in the serum of antibodies to Ro(SSA) or La(SSB) antigens, or both	Positive serum anti-SSA (Ro) and/or anti-SSB (La) or positive rheumatoid factor and ANA ≥ 1:320 Labial salivary gland biopsy exhibiting focal lymphocytic sialadenitis with a focus score ≥ 1 focus/4 mm^2^ Keratoconjunctivitis sicca with ocular staining score ≥ 3	The classification of SS applies to any individual who meets the inclusion criteria, does not have any condition listed as exclusion criteria, and who has a score ≥ 4 when summing the weights from the following items: Labial salivary gland with focal lymphocytic sialadenitis and focus score ≥1.3 → 3 weights Anti-SSA (Ro) → 3 weights Ocular staining score ≥ 5 (or van Bijsterfeld score ≥ 4) on at least one eye → 1 weight Schirmer ≤ 5 mm/5 min on at least one eye → 1 weight Unstimulated whole saliva flow rate ≤ 0.1 mL/min → 1 weight
Inclusion criteria	For primary SS a. The presence of any 4 of the 6 items is indicative of primary SS, as long as either item IV or VI is positive b. The presence of any 3 of the 4 objective criteria items (that is, items III, IV, V, VI) c. The classification tree procedure represents a valid alternative method for classification, although it should be more properly used in a clinical-epidemiological survey For secondary SS In patients with a potentially associated disease (for instance, another well-defined connective tissue disease), the presence of item I or item II plus any 2 from among items III, IV, and V		≥1 symptom of ocular or oral dryness (defined as a positive response to at least one of the following questions: (1) Have you had daily, persistent, troublesome dry eyes for more than 3 months? (2) Do you have a recurrent sensation of sand or gravel in the eyes? (3) Do you use tear substitutes more than 3 times a day? (4) Have you had a daily feeling of dry mouth for more than 3 months? (5) Do you frequently drink liquids to aid in swallowing dry food?), or suspicion of SS from the ESSDAI questionnaire (at least one domain with positive item)
Exclusion criteria	Past head and neck radiation treatment Hepatitis C infection Acquired immunodeficiency disease (AIDS) Pre-existing lymphoma Sarcoidosis Graft versus host disease Use of anticholinergic drugs (since a time shorter than 4-fold the half-life of the drug)	History of head and neck radiation treatment Hepatitis C infection AIDS Sarcoidosis Amyloidosis Graft versus host disease IgG4-related disease	History of head and neck radiation treatment Active Hepatitis C infection (with positive PCR) AIDS Sarcoidosis Amyloidosis Graft versus host disease IgG4-related disease

ESSDAI, EULAR Sjögren’s Syndrome Disease Activity Index; IgG, immunoglobulin G; PCR, polymerase chain reaction.

**Table 2 ijms-22-12903-t002:** Salivary biomarkers identified in SS.

Salivary Biomarker	Authors [Ref.]	Subjects	Sample	Used Criteria	Analytical Methods	Findings
β2-microglobulin	Markusse et al. [28]	39 pSS, 42 non-SS, 41 HC	Stimulated parotid saliva	-	ELISA	58% of pSS higher levels of mean + 2SD of levels of HC (7%)
	van der Geest et al. [29]	29 pSS, 30 HC	Unstimulated and stimulated parotid saliva	-	Radioimmunoassay	Higher in pSS (*p* < 0.001)
	Mogi et al. [31]	pSS, HC, sialoadenitis, diabetes mellitus	Unstimulated whole saliva	-	ELISA	Higher in pSS (*p* < 0.001)
	Ryu et al. [12]	41 pSS, 15 non-SS sicca, 20 HCs	Stimulated parotid saliva	2002 AECG	ELISA	Higher 4.3- and 3.7-fold for the low/medium and for the medium/high focus
	Hu et al. [15]	34 pSS, 34 SLE, 34 HC	Stimulated whole saliva	2002 AECG	ELISA	Higher in pSS (*p* = 1.25 × 10^−10^), AUC 0.87
	Baldini et al. [16]	19 pSS, 10 non-SS sicca SD, 25 sSS, 10 HC	Unstimulated whole saliva	2002 AECG	ELISA	Higher in pSS than HC (*p* < 0.001) and RA-sSS (*p* < 0.05)
	Asashima et al. [32]	71 pSS, 50 sSS, 54 non-SS-CTD, 75 HC	Unstimulated whole saliva	2002 AECG	ELISA	Higher in pSS (5.3 ± 4.6 mg/L) than non-SS-CTD (2.5 ± 2.1) and HC (1.2 ± 0.7)
	Garza-García et al. [33]	71 pSS	Unstimulated whole saliva	2012 SICCA	ELISA	Positive correlation with ESSPRI (Kendall’s tau 0.759, 95% CI 0.656–0.837, *p* < 0.0001)
Lactoferrin	Ryu et al. [12]	41 pSS, 15 non-SS sicca, 20 HCs	Stimulated parotid saliva	2002 AECG	ELISA	Higher 3.7- and 3.6-fold in pSS patients with low/medium focus and medium/high focus
	Markusse et al. [28]	39 pSS, 42 non SS sicca SD, 41 HC	Stimulated parotid saliva	-	ELISA	26% of pSS higher levels of mean + 2 SD of levels of HC (0%)
	Konttinen et al. [35]	3 pSS, 5 sSS, 8 HCs	Unstimulated whole saliva	-	radioimmunoassay	Higher in SS (48.92 ± 14.21 μg/mL) than HC (4.03 ± 1.48)
NGAL	Aqrawi et al. [42]	11 pSS, 11 HCs	Stimulated whole saliva	2002 AECG	LC-MS	Detected in 8/11 pSS and 2/11 HCs
Siglec-5	Lee et al. [45]	170 pSS, 43 SLE, 25 HCs	Unstimulated whole saliva	2012 SICCA	WB, ELISA	Higher in pSS than non-SS (*p* < 0.001) Negative correlation with salivary flow rate, positive correlation with ocular surface score and serum immunoglobulin G, AUC 0.774
TRIM29	Sembler-Møller et al. [23]	24 pSS, 16 non-SS	Unstimulated and stimulated whole saliva	2016 ACR/EULAR	nano-scale LC-MS	AUC 0.881 for pSS
Proinflammatory cytokines	Nguyen et al. [46]	21 pSS, 19 HCs	Unstimulated whole saliva	2002 AECG	ELISA	No difference in levels of IL-17 Higher levels of IL-6 in pSS (*p* < 0.05)
	Benchabane et al. [47]	44 pSS, 15 HCs	Stimulated whole saliva	2002 AECG	ELISA	Higher IL-17A, IL-6, TNF-α, IL-10 in pSS
	Kabeerdoss et al. [48]	43 pSS, 31 HCs	Unstimulated whole saliva	2002 AECG or 2012 SICCA	ELISA	Higher median levels of sL-selectin, IL-7 in pSS No association between ESSDAI and levels of sL-selectin, IL7
	Hung et al. [49]	138 pSS, 100 HCs	Unstimulated whole saliva	2002 AECG	ELISA	Higher IL-6 (*p* = 0.012), but not TNF-α, IL-17A, RF-IgA IL-6 correlated with ESR (r = 0.252), IgG (0.248)
	Tvarijonaviciute et al. [60]	17 SS, 13 HCs, 19 non-SS sicca	Unstimulated whole saliva	2002 AECG	ELISA	Higher IL-1β vs. HC or non-SS sicca (both *p* < 0.001) Higher IL-8 vs. HC or non-SS sicca (both *p* < 0.001)
Anti-histone, anti-transglutaminase, anti-SSA, and anti-SSB Ab	Hu et al. [50]	34 pSS, 34 SLE, 34 HCs	Unstimulated whole saliva	2002 AECG	ELISA	AUC of anti-histone, anti-transglutaminase, anti-SSA, anti-SSB antibody 0.95, 0.87, 0.93, 0.94
Muscarinic type 3 receptor (M3R)	Jayakanthan et al. [52]	43 pSS, 34 SLE, 42 HCs	Unstimulated whole saliva	2002 AECG or 2012 SICCA	ELISA	Positive in 55.81, 17.64, 7% of pSS, SLE, HCs Positivity was associated with lower age, shorter disease duration, higher globulin levels
	Mona et al. [53]	37 pSS, 26 non SS-sicca	Unstimulated and stimulated whole saliva	2002 AECG	On-Cell-Western assay	Higher (3.59) in pSS, AUC 0.84
Tissue-specific antibodies (anti-CA6, SP1 and PSP)	Jin et al. [54]	137 pSS, 32 SLE-sSS, 127 HCs	Unstimulated whole saliva	2002 AECG or 2012 SICCA	ELISA	Higher anti-CA6 IgG (*p* < 0.01), anti-SP1 IgG (*p* < 0.01), anti-PSP2 IgG (*p* < 0.05) in pSS than HCs
Calprotectin	Jazzar et al. [59]	51 SS, 14 SS with MALT lymphoma, 18 HCs	Unstimulated whole and parotid saliva	2002 AECG and 2012 SICCA	ELISA	Higher S100A8/A9 in parotid saliva of SS (743.1 ng/mL) than HC (31.9, *p* = 0.001) and than SNOX (208.9, *p* = 0.031)
Adiponectin	Tvarijonaviciute et al. [60]	17 SS, 13 HCs, 19 non-SS sicca	Unstimulated whole saliva	2002 AECG	ELISA	Higher adiponectin in SS than HC (*p* = 0.034) and non-SS sicca (*p* = 0.007) Higher ADA in SS than HC (*p* = 0.034) and non-SS sicca (*p* = 0.033)

pSS, primary Sjogren’s syndrome; AECG, the American-European Consensus Group criteria; SICCA, the American College of Rheumatology/Sjögren’s International Collaborative Clinical Alliance criteria; ACR/EULAR, the American College of Rheumatology/European league against rheumatism classification criteria; NGAL, neutrophil gelatinase-associated lipocalin; siglec, soluble sialic acid-binding immunoglobulin-like lectin; TRIM, tripartite motif-containing protein; CA6, carbonic anhydrase VI; SP, salivary protein; PSP, parotid secretory protein; SLE, systemic lupus erythematosus; HC, healthy controls; ELISA, Enzyme-linked immunosorbent assay; SD, standard deviations; AUC, area under curve; IL, interleukin, TNF, tumor necrosis factor; ESSPRI, The EULAR Sjögren’s syndrome Patient Reported Index; ESSDAI, EULAR Sjögren’s Syndrome Disease Activity Index; sL-selectin, soluble L-selectin; RF, rheumatoid factor; SNOX, sialadenitis, nodular osteoarthritis and xerostomia; ADA, adenosine deaminase.

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
