# Peer review of "Salivary Biomarkers in Patients with Sjögren’s Syndrome—A Systematic Review"

_ijms, 2021, doi:10.3390/ijms222312903_

Round 1

Reviewer 1 Report

Type of manuscript: Review

Title: Salivary biomarkers in patients with Sjögren’s syndrome—a Systematic Review

Ju-Yang Jung, Ji-won Kim, Hyoun-Ah Kim, and Chang-Hee Suh

Journal: International Journal of Molecular Sciences

In this review the authors report a research carried out in the scientific literature on the presence of markers contained in the saliva of SS patients who show a significantly different level of expression compared to healthy subjects.

Although the topic may be interesting and useful for identifying new therapeutic targets, however, the organization of the manuscript and its contents are too simplistic. I suggest the authors to investigate the individual markers by providing more information on the activities in which these markers are involved and by reporting the molecular mechanisms of which they are actors. This must be done for each of the examined markers. the authors must also underline that those examined in the literature are only some of the factors contained in saliva that exhibit a different expression compared to healthy subjects.

Author Response

Dear Reviewer,

We appreciate your review of our manuscript “Salivary biomarkers in patients with Sjögren’s syndrome—a Systematic Review". In response to your comments, we have made several changes and added the necessary clarifications, as summarized below:

  1. Although the topic may be interesting and useful for identifying new therapeutic targets, however, the organization of the manuscript and its contents are too simplistic. I suggest the authors to investigate the individual markers by providing more information on the activities in which these markers are involved and by reporting the molecular mechanisms of which they are actors. This must be done for each of the examined markers.

Answer) Thank you for the comment. We added the molecular mechanisms of the markers and associations to characteristics of Sjögren’s syndrome, and the change is underlined in the revised version of the manuscript.

  1. The authors must also underline that those examined in the literature are only some of the factors contained in saliva that exhibit a different expression compared to healthy subjects.

Answer) Thank you for the commend. We added and the change is underlined in the revised version of the manuscript.

We thank you for the constructive review and hope that the revised manuscript now meets the journal's standards for publication.

Reviewer 2 Report

This is a detailed review on biomarkers related to SS.

I have just one question:

There are several classification criteria for SS (2002 AECG, 2012 SICCA, 2016 ACR/EULAR). In this review artcle, cited references vary from old (1990s) to newest (2021). Did the authors take classification criteria into consideration for making tables?

Author Response

Dear Reviewer,

We appreciate your review of our manuscript “Salivary biomarkers in patients with Sjögren’s syndrome—a Systematic Review". In response to your comments, we have made several changes and added the necessary clarifications, as summarized below:

There are several classification criteria for SS (2002 AECG, 2012 SICCA, 2016 ACR/EULAR). In this review article, cited references vary from old (1990s) to newest (2021). Did the authors take classification criteria into consideration for making tables?

Answer) Thank you for the comment. We added Table 1 and classification criteria information in Table 2 in the revised version of the manuscript.

We thank you for the constructive review and hope that the revised manuscript now meets the journal's standards for publication.

Round 2

Reviewer 1 Report

the manuscript was improved after revision providing interesting scientific information to identify potential markers of SS